# Equilibrium and Kinetic Modeling of Crystal Violet Dye Adsorption by a Marine Diatom, *Skeletonema costatum*

**DOI:** 10.3390/ma15186375

**Published:** 2022-09-14

**Authors:** Mohamed Ashour, Ahmed E. Alprol, Mohamed Khedawy, Khamael M. Abualnaja, Abdallah Tageldein Mansour

**Affiliations:** 1National Institute of Oceanography and Fisheries (NIOF), Cairo 11516, Egypt; 2Department of Chemistry, College of Science, Taif University, P.O. Box 11099, Taif 21944, Makkah, Saudi Arabia; 3Animal and Fish Production Department, College of Agricultural and Food Sciences, King Faisal University, P.O. Box 420, Hofuf 31982, Al-Ahsa, Saudi Arabia; 4Fish and Animal Production Department, Faculty of Agriculture (Saba Basha), Alexandria University, Alexandria 21531, Egypt

**Keywords:** algae, bioremediation, crystal violet dye, modeling, adsorption kinetics

## Abstract

Significant efforts have been made to improve adsorbents capable of eliminating pollutants from aqueous solutions, making it simple and quick to separate from the treated solution. In the current study, the removal of Crystal Violet Dye (CVD) from an aqueous synthetic solution onto a marine diatom alga, *Skeletonema costatum,* was investigated. Different experiments were conducted as a function of different pH, contact time, adsorbent dosage, temperature, and initial CVD concentration. The highest adsorption efficiency (98%) was obtained at 0.4 g of *S. costatum*, pH 3, and a contact time of 120 min, at 25 °C. Furthermore, Fourier-transform infrared spectroscopy (FTIR) results display that binding of CVD on *S. costatum* may occur by electrostatic and complexation reactions. Moreover, the Brunauer–Emmett–Teller surface area analysis (BET) obtained was 87.17 m^2^ g^−1^, which, in addition to a scanning electron microscope (SEM), reveals large pores that could enhance the uptake of large molecules. However, the equilibrium adsorption models were conducted by Halsey, Langmuir, Freundlich, Henderson, and Tempkin isotherm. In addition, multilayer adsorption isotherm best described the uptake of CVD onto *S. costatum*. The maximum monolayer adsorption capacity (q_max_) was 6.410 mg g^−1^. Moreover, thermodynamic parameters of the adsorption studies suggested that the uptake of CVD onto *S. costatum* was endothermic and spontaneous. The pseudo-first-order, pseudo-second-order, and intra-particle diffusion kinetic equations were applied to model the adsorption kinetic data. It was seen that the kinetics of the adsorption may be described using pseudo-second-order kinetic equations. Finally, the present work concluded that the marine diatom alga *S. costatum* is suitable as a natural material for the adsorption of CVD.

## 1. Introduction

Water is a necessary component of life and is utilized by almost all living things on the world’s surface [1]. It is also a valuable resource in world economies because agriculture and other commercial uses consume around 70% of freshwater [2,3]. The untreated discharge of synthetic dyes from the industrial sector (textiles, printing, dyeing cosmetics, paints, plastic, rubber, leather, food, etc.) is a major concern for researchers working on wastewater pollution protection [4,5]. Moreover, industrial wastewater mainly contains several organic complicated molecules and dyestuff which are discharged into highly dyed wastewater, which has caused critical environmental concerns globally [6,7,8,9,10,11,12]. Intensive technological and industrial development, along with the increase in other human activities, has led to the overuse of water resources and increased water pollution [7,13]. The textile industry is the main industry that utilizes dyes causing the main source of water pollution, which poses a threat to aquatic life and human life [14,15]. Therefore, dye removal from wastewater poses a challenge to prevent dyes from causing harm to the environment [8], especially to aquatic biological systems [16]. To manage the problems of water pollution, numerous physical, chemical, and biological approaches to decomposing many textile dyes were used over the latest years [8]. For example, advanced oxidation processes, nanofiltration, photocatalytic degradation, adsorption, and biological treatment are among the technologies used to remove organic pollutants [2,17,18,19]. The primary objective of these strategies was to reduce damaging dyes into non-toxic end products, but most of them failed, since the end compounds were more toxic than the primary dyes [20,21]. Among these technologies, adsorption has grown as an effective biological and economical technique for decontamination of organic pollutants from wastewater [22,23,24,25,26]. Sharma et al. [26] reported that adsorption is considered a potential technique because of its ease of operation and relatively low cost of application required for wastewater pollution removal.

There has been a lot of attention recently to converting marine biomass applications into value-added products [27,28,29], especially sorbents, to reduce the impacts of environmental pollutants [23]. Recently, algal cells (microalgae and/or seaweeds) are one of the most important low-cost, eco-friendly, sustainable, and biologically promising technologies for dye and pollutant removal from aqueous solutions [8,30,31]. Algal cells have a long history of bioremediations of toxic dyes and/or water pollutants [32,33,34]. Generally, due to their biomolecules, bioactive compounds, and their functional groups, algae are the richest organisms in our world that can serve and be utilized in several bio-industries, such as human food supplements, animal feeds, bio-pharmaceuticals, bio-cosmetics, bio-fertilizers, fine-biochemicals, bio-energy, and bio-plastic, etc. [35,36,37,38,39,40,41,42,43,44,45,46]. Microalgae have a great prospect of interacting with water pollutants as a major producer in the food chain and a major player in the balance of the marine ecosystem [47,48,49]. Additionally, the use of dead organisms in adsorption is more beneficial for the treatment of water, since dead microorganisms are not affected by harmful wastes, do not necessitate a constant supply of nutrients, and can regenerate and be reused numerous times [50]. Dead cells can be kept or used for a long time at room temperature without putrefaction taking place. Additionally, it has been demonstrated that dead cells acquire contaminants to an equal or greater degree than developing or resting cells [51]. The use of dead biomass in powdered form in the column does, however, present several challenges, including the difficulty of easily separating the biomass after adsorption, mass loss after regeneration, low strength and density, and small particle size, which make it challenging to utilize them in column applications. Dead biomass can be immobilized in supportive material to address these issues [52].

The marine diatom *Skeletonema costatum* is a chain of microalgal cells extensively used in marine hatcheries as live feed for marine larvae [53,54,55]. In addition, *S. costatum* species have many bioremediation advantages in adsorbing several pollutants from the aquatic environment, in a process that is simple, easy to observe, and sensitive to toxins [56,57]. The bioremediation process of *S. costatum* is carried out via passive transport, through the cell wall, and an active transport mechanism [3,58]. The *S. costatum* cell wall contains numerous functional groups, such as amine, hydroxyl, carboxyl, phosphate, and thiol groups. These functional groups, mainly negatively charged, can react with the positively charged ions of many pollutants [56,59]. The dye removal, especially by using algae, may be attributed to the accumulation of dye ions on the surface of algal biopolymers and further to the diffusion of the dye molecules from the aqueous phase to the solid phase of the biopolymer [60]. Extracellular polymers consist of surface functional groups, which enhance the sorption of the dye molecules onto the surface of the polymer (floc) during the dye removal process [61].

Crystal violet dye (CVD) is mainly used during the manufacturing process to dye cotton, silk, inks, and paints [62]. The protein CVD is used to detect bloody fingerprints. CVD is also applied throughout many kinds of adhesive taps for latent printing. Gram’s stain is frequently employed in the demonstration and basic classification of dyes [22]. CVD is also used to create vibrant dyes [18,63]. It is carcinogenic and has been classified as a recalcitrant molecule, since it is poorly metabolized by microbes, it is non-biodegradable, and it can persist in a variety of environments [64]. Furthermore, biological waste materials, eco-friendly behavior, and the possibility of reusability or regenerating to find low-cost adsorbent materials and reactivity, thus acting as suitable adsorbents, have been considered [65,66].

The objective of the present research was to explore the possibility of using the marine diatom alga *Skeletonema costatum* as an adsorbent for dye elimination, for example, CVD, a widely used dye in the textile-processing industry. The impact of experimental parameters such as pH, contact times, adsorbent dosages, temperatures, and initial dye concentrations on the adsorptive capacity of the adsorbent was examined. On the other hand, thermodynamic properties, equilibrium adsorption isotherms, and adsorption kinetics were examined by using conventional theoretical methods to investigate the adsorption mechanism. Moreover, the industrial wastewater and seawater applications, regeneration, and reusability have been studied. Therefore, in this study, this material was selected to investigate its ability to remove toxic dyes. However, even though there is literature on the marine diatom alga *Skeletonema costatum* as an adsorbent for the removal of heavy metal ions, where the previous studies showed the insufficiency of the use of this material in removing toxic dyes from wastewater, this work shows an experimental study on the removal of dyes, especially CVD, by using this novel material. Since it considerably improved the adsorption effectiveness of the specified dyes, it was chosen for the following investigation to cover the gap in the literature. Therefore, the current work aims to create a high-performance, environmentally friendly powder adsorbent from *Skeletonema costatum* culture for the removal of CVD from an aqueous solution. Furthermore, *Skeletonema costatum* was considered due to its novelty, which may easily be regenerated and recycled, which makes it applicable for treatment processes. Moreover, the highest suitable model was achieved by using error functions.

## 2. Materials and Methods

### 2.1. Materials

#### 2.1.1. *Skeletonema costatum*

The marine diatom alga *Skeletonema costatum* was identified and isolated previously from the Eastern Harbor of Alexandria Coast (31°13′48″ N; 29°53′12″ E), close to the National Institute of Oceanography and Fisheries (NIOF), Egypt. An isolated diatom strain, *S. costatum* (cell size from 2 to 5 μm), was cultured at an indoor scale under standard conditions of temperature (22 ± 2 °C), salinity (34 ± 1 ppt), continuous illumination (3500 ± 500 Lux), and continuous aeration. After a batch culture in conical flasks (1 L) filled with F/2 standard Guillard medium [67], the culture cells were harvested by centrifuge (3000× *g*. 10 min^−1^) at the late exponential phase on day 5. The harvested biomasses were dried (at 55 °C for 48 h) and preserved at −20 °C for further applications.

#### 2.1.2. Crystal Violet Dye (CVD)

The adsorbate, crystal violet (CV) dye, was a basic, acquired from Daystar, Mumbai, India (basic dye, C.I. 42555, max. abs. = 588 nm, molecular formula C_25_H_30_N_3_Cl and standard purity (97.5% specified)) (Figure 1). Distilled water was used to obtain the desired concentrations. The original pH of the dye solution was 2.0. The maximum absorbance of the dye solutions was monitored using a spectrophotometer to determine their concentrations (Milton Roy, spectronic 2ID).

#### 2.1.3. Chemical and Reagents

The chemicals used in the adsorption study, including sodium hydroxide (NaOH, 98.6%) and sulfuric acid (H_2_SO_4_, 98%), were obtained from Fisher Scientific (Fisher Scientific, Montreal, QC, Canada). All the applied chemicals throughout this study were analytical-grade reagents.

### 2.2. Methods

#### Batch Adsorption Process

Batch adsorption trials were done in a shaker at various ranges of parametric conditions. Optimization reactions were achieved for pH factors in the range 3, 5, 7, 9, and 11, initial CVD concentration influence in the range 5, 10, 15, 20, and 40 mg L^−1^, the contact time parameter (15, 30, 60, 120, and 180 min), biomass amount (0.05, 0.1, 0.2, 0.3, and 0.4 g), and temperature influence at 25, 35, 45, and 55 °C. For these aims, the used bottles were shaken at a controlled temperature (25 °C), using an orbital shaker (Orbital incubator model SI50, Stuart Scientific, Redhill, UK) for the minimal contact time necessary to achieve equilibrium, as indicated by the kinetic data, with an agitation rate of 110 rpm.

The efficiency of the treatment was assessed to determine the concentration using the UV–visible spectroscopic method at 588 nm (Milton Roy, Spectronic 21D, Houston, TX, USA). The impact of pH was evaluated by modifying the reaction mixture to various initial values of pH and evaluating the remaining color at the equilibrium contact time. Liquid solutions of NaOH and H_2_SO_4_ were utilized to regulate the needed values of pH. The percentages of removal and q_e_ (mg g^−1^) of CVD onto *S. costatum* were calculated, as described by Ghoneim, et al. [68], as the following:(1)$$q_{e\;}=\frac{\left(C_{i}-C_{f}\right)\times V}{W}$$
(2)$$\mathrm{Percentage}\;\mathrm{removal}\;\left(\%\right)=\frac{\left(C_{i}-C_{f}\right)}{C_{i}}\times 100$$
where C_i_ and C_f_ are the initial and the final CVD (mg L^−1^), respectively; W is the quantities of *S. costatum* adsorbent (g); and V is the solution volume (L).

For kinetic observations, 10 mL flasks containing CVD (5 mg L^−1^) and the adsorbent (0.05 g) were exposed to agitation for numerous contact time intervals. At regular intervals, the flasks were removed from the shakers, and the remaining concentration of CVD solution was calculated.

### 2.3. Characterizations

Before and after the adsorption process, the surface morphology of the adsorbents was characterized via a Scanning Electron Microscope, SEM (JSM-IT200 with an accelerating voltage of 20.0 kV at a magnification ×2000). The samples were coated with platinum to prevent charging. On the other hand, the functional groups of the adsorbents, as well as a binding mechanism (s), were conducted using a Fourier-transform infrared spectrometer (FTIR) (Thermo Nicolet 6700). Furthermore, in this study, quantachrome instruments (cyl. pore) (NLDFT Ads. model) were used to examine the surface area of the raw adsorbent at 77.35 K.

### 2.4. Data Analysis

The normality and homogeneity assumptions were performed before the statistical analysis of the data. The experiments were applied in triplicates and the obtained data were presented in the form of means ± standard deviation (SD, *n = 3*). Before analysis, all results in percentages were arcsine transformed [69].

## 3. Results and Discussion

### 3.1. S. costatum (Adsorbent) Characterization

#### 3.1.1. SEM

The scanning electron microscope investigations of *S. costatum*, before and after the adsorption process (Figure 2), concluded that the adsorbent (Figure 2A) has an irregular and heterogeneous surface morphology with a well-developed porous structure. Moreover, pores of several sizes and shapes were determined. The presence of these fine particles leads to an increase in the surface area and the porosity of *S. costatum* [70]. After adsorption of CVD (Figure 2B), the remarkable changes were investigated in the morphological structure of CVD-loaded *S. costatum*, revealing an irregular surface with a molecular cloud of the CVD, where the CVD may be adsorbed and trapped. The higher the porosity, the higher the adsorption capacity [17].

#### 3.1.2. FTIR

FTIR spectra of the *S. costatum* before and after the adsorption process confirmed that there are several functional groups, as shown in Figure 3.

The wide peaks, which are located at around 3403, 3295, and 3495 cm^−1^, are typically due to adsorbed water and/or the groups of hydroxyl [64]. The bands that existed around 2922 and 2856 cm^−1^ correspond to C–H vibrations of methyl and methylene groups. However, the band around 1645 and 1662 cm^−1^ generally happened by the stretching vibration of C=O carboxyl groups, while the bands around 1548 and 1542 cm^−1^ are ascribed to the aromatic ring or C=C stretching vibration [71]. In addition, the bands at around 1406 and 1458 cm^−1^ correspond to the C–H in-plane bending vibrations in methylene and methyl groups. The band at 1227–1230 cm^−1^ corresponds to C–C stretching in alkanes and a comparatively intense band at about 1081 and 1081 cm^−1^ can be assigned to C–O stretching vibrations in alcohols. The C–H out-of-plane bending vibrations cause the bands at 853 and 871 cm^−1^. Disappearance and band shifting from 1406 to 1457 cm^−1^ are determined in the CVD-adsorbed *S. costatum* attributed to the stretching of C=C modes of the benzene ring of lignin. Transmittance at wave number 3429 cm^−1^ is found to be shifted to 3403 cm^−1^ on adsorption and this perhaps assumes the responsibility for the chemical reaction of the CVD with O–H groups on the adsorbent. Several bands were found in the region of 1000–500 cm^−1^, which is perhaps attributed to the C–H stretching. From FTIR observations, CV dye ions may interact with the active surface of *S. costatum* adsorbents, resulting in the creation of new absorption bands, changes in absorption strength, and a movement in the wavenumber of functional groups. CVD ions bound to the active sites of the *S. costatum* adsorbents through complexation mechanism and electrostatic attraction. In addition, the electrostatic attraction was between carbonate groups and CVD ions. Furthermore, the complexation mechanism engaged in sharing the electron pair between electron donor atoms (N and O). The current work determined that carbonate, carbonyl, hydroxyl, and amine are the primary adsorption sites in *S. costatum* adsorbents. Another study found that Cu adsorption by *S. costatum* has a high percentage of removal due to the same functional groups being responsible for the adsorption process, which were mainly carboxyl, hydroxyl, and amine, as well as, thiol and phosphate groups, as reported by Pratama et al. [56,57,59].

#### 3.1.3. BET

Table 1 shows the surface characteristics of the dry adsorbent employed in this investigation based on adsorbent-specific surface area characterization using the BET technique, according to nitrogen adsorption and desorption isotherms. Table 1 and Figure 4 show the adsorbent’s surface area was 87.172 m^2^ g^−1^, as determined by the results. The total pore volume and average pore size of *S. costatum* were also evaluated and found to be 0.103 CC g^−1^ and 3.131 nm, respectively.

### 3.2. Adsorption Studies

#### 3.2.1. pH

The pH value is an important parameter affecting the removal uptake of toxic pollutants from wastewater due to the variation of H^+^ and OH^−^ in the treatment medium [72]. The effect of solution pH on adsorption of CVD on *S. costatum* was studied by varying the pH of the dye solution for an initial concentration of 5 mg L^−1^, while the adsorbent dosage, contact time, agitation speed, and temperature were fixed at 0.4 g/50 mL, 180 min, 110 rpm, and 25 °C, respectively. The obtained results indicate that the adsorption process of CVD was favorable in an acidic medium, as shown in Figure 5.

Dye removal efficiency decreased when the pH increased from 5 to 11, indicating that pH did not significantly affect the percentage of CVD removal, particularly under alkaline conditions. The highest percentage of CVD uptake was obtained at pH 3, with a percentage removal of 96%. CVD is based on chromophores combined with several kinds of reactive groups that react with the active functional groups of *S. costatum*, such as chitin, carboxyl, hydroxyl, acidic polysaccharides, amine, and the other functional groups [57,59]. Higher removal observed at low pH values is perhaps attributed to the electrostatic appeal between negatively charged CVD anions and positively charged *S. costatum* cell surface [73]. Furthermore, as the pH value increases, the adsorption of CVD decreases because of competition between the anionic dye and excess OH^−^ ions in the solution [74]. According to Wu et al. [75], the amount of adsorption decreased because the negatively charged adsorbent gave fewer effective sites for adsorption as repulsive forces increased. Following that, adsorption experiments were carried out at the optimal pH.

#### 3.2.2. Adsorbent Dose of *S. costatum*

The effect of different *S. costatum* levels (0.05, 0.1, 0.2, 0.3, and 0.4 g/50 mL) were studied at 25 °C, for 180 min. With the increasing doses of *S. costatum*, the quantities of CVD adsorption and the percentage of removal increased from 90.90% to 95.20% and from 0.05 to 0.4g (Figure 6). This phenomenon can be attributed to an increase in the adsorbent-specific surface area and the availability of more adsorption sites [76]. Furthermore, this pattern might be described as a result of partial biomass agglomeration at increasing biomass concentrations, resulting in a reduction in the specific surface area of adsorption [77]. Only about a 3% increase was observed as the adsorbent dose was increased from 0.05 to 0.03 g; overlap of adsorption sites may describe the small increase in percentage adsorption. The decrease in the amount of CVD adsorbed q_e_ (mg g^−1^) with increased adsorbent mass is attributed to a division of the flow or a concentration gradient between the solution and the absorbing surface. When *S. costatum* sorbent mass increases, the level of CVD that is adsorbed onto the unit weight of *S. costatum* decreases, resulting in a drop in the q_e_ value as the adsorbent mass concentration rises [78]. In the present study, the adsorbent dose of 0.4 g/50 mL is the optimal value for the removal of CVD.

#### 3.2.3. CVD Concentrations

The effect of initial CVD levels (5, 10, 15, 20, and 40 mg L^−1^) on adsorption onto 0.1 g/ 50 mL of the dried *S. costatum* for a contact time of 3 h at 25 °C was investigated. Figure 7 shows that the uptake capacity and adsorption capacity of CVD by *S. costatum* increased with the increasing initial level of CVD. This increase in the amount of adsorption with increased initial CVD concentration may be due to the increased driving force that dominates the resistance of mass transfer of the CVD between the solid phase and bulk liquid phase [79].

Furthermore, the surface of the sorbent is bare in the initial stage; sorption happens quickly and is usually regulated by the diffusion process from the bulk to the surface. Since there are fewer sorption sites accessible at this stage, the sorption is most likely an attachment-controlled process. Increasing the initial concentration results in increased diffusion of more CVD molecules in the bulk solution towards the external surface of the *S. costatum* and enhanced interaction between crystal violet anions and the *S. costatum* surface leading to increased adsorption uptake of CVD. The adsorption typically starts rapidly and then gradually slows because when a significant number of vacant surface sites seem to have been able to adsorb at first, then it is always difficult to occupy the residual vacant surface sites due to repulsive forces between the CVD molecules on the *S. costatum* and the bulk phase [74]. The obtained data indicate that the efficiency of *S. costatum* in the removal of CVD was highly related to the initial concentration of CVD. Similar suggestions were reported by Alprol et al. [80], who indicated that the efficiency of *A. platensis* in the removal of IV2R was highly related to the initial concentration of IV2R. Inventor et al. [81] confirmed that initial quick adsorption is due to dye molecules contacting available surface adsorption sites, while the following gradual adsorption is due to dye molecules’ absorption into the pores of the adsorbents.

#### 3.2.4. Contact Times

The practical efficiency of the adsorption technique in solution is largely based on the immersion time in the medium (i.e., contact time). Figure 8 illustrates the amount of CVD adsorbed onto *S. costatum* as a time function at 0.1 g/50 mL. CVD adsorptions were fast initially and gradually reduced over time until equilibrium was attained. Fast adsorption was observed within the first 15 min (87.88%) and then the adsorption rate of dye is high, which may be attributed to abundant adsorption sites, which attracted the dye particles from the bulk solution at the initial stages. The adsorption process becomes less efficient due to the gradual occupancy of these sites. In the figure, it is shown that the adsorption reaction nearly attained equilibrium within 120 min, after which no substantial amount of crystal violet was adsorbed with increasing contact time after equilibrium was achieved, indicating that all available sorption sites had been occupied [68]. Due to increased repulsive forces between dye molecules and the bulk solution, the occupation of the remaining unoccupied spots becomes increasingly difficult as time passes.

#### 3.2.5. Temperature

The temperatures of 25, 35, 45, and 55 °C were studied with 0.1/ 50 mL (Figure 9). The maximum percentage removal (79.30%) of CVD was observed at 25 °C. Based on the relation between removal efficiency and temperature, the results reported that the CVD sorption takes place through the aggregation of the dimers and/or monomers of CVD. The dimeric or monomeric dispersed from the cellular membrane are perhaps trapped as an outcome of accumulation, even at a low equilibrium CVD level. The relative increase in dye level in the cell probably leads to aggregation. Aggregation is most likely caused by a relative rise in CVD concentration in the cell. Hence, the reduction of sorbed quantities at higher temperatures, perhaps due to deaggregation, is based on the relative increase in thermal motions of CVD aggregates [82].

### 3.3. Thermodynamics Properties

To investigate the thermodynamic properties and thermal impact on the adsorption process, several temperatures (25, 35, 45, and 55 °C) were conducted with 0.05 g/50 mL of *S. costatum*. Different parameters of thermodynamics such as entropy change (ΔS), enthalpy change (ΔH), and Gibbs free energy (ΔG) were calculated to determine the natural process of the adsorption, as shown in Table 2.

The magnitude of ΔG (kJ mol^−1^) was conducted using the following equation:(3)∆G=− RTLnKa
where R is the universal gas constant (0.008314 kJ mol^−1^ K); T is the absolute temperature (°K); and Ka is the adsorption equilibrium constant. ΔH (kJ mol^−1^) was calculated by the following equation:(4)∆H= ΔG +TΔS 

A plot of ∆G against T was found to be linear and the value of ∆H and ∆S was calculated from the slope and intercept of the plots. The negative values of ∆G for *S. costatum* have confirmed the feasibility of the process and the spontaneous nature of adsorption. However, the values of ∆G were increased from −5.050 to −2.636 kJ mol^−1^ as the temperature increased from 25 to 55 °C. The values of ∆H and ∆S were calculated and are given as 8.636 kJ mol^−1^ and −0.036 kJ mol^−1^ K, respectively. Hossain et al. [83] and Subbareddy et al. [84] reported that the positive value of ∆H indicates that the adsorption reaction was endothermic, and the strong affinity of the *S. costatum* diatom for CVD ions suggests some structural changes in CVD ions and the *S. costatum* [85]. In addition, the negative value of ∆S also suggests that the adsorption was enthalpy-driven and spontaneous [86]. Furthermore, the negative ∆S reflected the affinity of the *S. costatum* for the CVD and the decrease of randomness at the solid/liquid interface occurring in the internal structure during the adsorption process [75,86]. The negative value of ∆S shows a decline in the degree of freedom of the adsorbed CVD ions [87].

### 3.4. Equilibrium Adsorption Isotherm

Adsorption isotherms are essential in calculating the maximum capacity and the kind of adsorption front that is produced. To characterize the equilibrium nature of *S. costatum*, various adsorption isotherms have been examined [82]. The data were fitted to the Freundlich and Halsey isotherm equations and the fixed parameters of the isotherm equations were calculated. 

#### 3.4.1. Langmuir Isotherm

The Langmuir model [88] assumes monolayer adsorption (the adsorbed layer is one molecule in thickness), with adsorption only occurring at a finite (fixed) number of definite localized sites that are identical and equivalent, with no lateral interaction and steric hindrance between the adsorbed molecules, even on adjacent sites. In its derivation, the Langmuir isotherm refers to homogeneous adsorption, in which each molecule possesses constant enthalpies and sorption activation energy (all sites possess an equal affinity for the adsorbate) [89]. The nonlinear form of the Langmuir equation was calculated by the following non-linear form (Lineweaver–Burk linearization of the Langmuir model—Type 2) [90]:q_e_ = q_max_ K_L_ C_e_/(1 + K_L_C_e_)(5)
where q_max_ (mg g^−1^) is the maximum sorption capacity corresponding to the saturation capacity (representing the total binding sites of an adsorbent) and K_L_ (L mg^−1^) is a coefficient related to the affinity among the adsorbent and dye ions.

The relationship was calculated from plotting curve (1/q_e_) vs. (1/C_e_):1/q_e_ = 1/(K_L_ q_max_ C_e_) + 1/q_max_(6)
where K_L_ and q_max_ are determined from the slope and intercept, respectively.

The CVD dye showed a maximal uptake capacity (q_max_) of 17.76 mg g^−1^ by the *S. costatum.* The observed linear correlation coefficient (R^2^ = 0.876) indicates this model does not give a good fit to the adsorption process (Figure 10). The Langmuir constant (K_L_), which is related to the heat of adsorption, was found to be 5.06. The Langmuir constant K_L_ can serve as an indicator of the strength or affinity of the sorbent for the solute [87].

The essential characteristics of the Langmuir isotherms can be expressed in terms of a dimensionless constant separation factor or equilibrium parameter, R_L_, which is defined by Hall et al. [91]:R_L_ = 1/(1+ K_L_ C_i_)(7)
where Ci is the initial adsorbate concentration (mg L^−1^). The R_L_ value indicates the shape of the isotherm, as shown in Table 3. R_L_ values lie between 0 and 1, which indicates a favorable sorption isotherm for CVD. The RL was found to be 0.011.

#### 3.4.2. Freundlich Isotherm

The Freundlich model can be applied to multilayer adsorption, with non-uniform distribution of adsorption heat and affinities over the heterogeneous surface [93]. Using the observed data of the *S. costatum* experiment, the feasibility of the empirical Freundlich isotherm was determined depending on sorption on heterogeneous surfaces, by plotting log (q_e_) versus log (C_e_). The Freundlich adsorption isotherm was expressed in the equation as the following [94]:(8)$$q_{e}=K_{f}\;C_{e}^{1/n}$$
Log q_e_ = Log K_F_ + 1/n LogC_e_
(9)

Freundlich constants K_F_ and 1/n were determined from the isotherm equation according to Equation (9). Freundlich adsorption isotherm constants and correlation coefficients are shown in Table 4. As its value is closer to zero, the constant 1/n, which represents the intensity of dye adsorption onto the sorbent or surface heterogeneity, becomes more heterogeneous. The n_F_ value reflects the degree of nonlinearity between solution concentration and adsorption in the following ways: if nF = 1, adsorption is linear; if nF < 1, the adsorption process is chemical; if nF > 1, the adsorption is a beneficial physical process [95]. The value of n was calculated to be 5.102, showing that the adsorption process is chemical. In addition, the value of 1/n (0.196) was found to lie between 0 and 1, indicating that CVD is confidently adsorbed by the *S. costatum* diatom adsorbent. Correspondingly, the obtained results fit with the data of the experiment of the Freundlich isotherm model, which was confirmed by a high correlation coefficient of R^2^ = 0.936 for the adsorbent (Figure 11). Therefore, the Freundlich isotherm can be employed to describe heterogeneous systems.

#### 3.4.3. Tempkin Isotherm

According to the Tempkin isotherm model, the heat of adsorption for every molecule in the layer falls linearly with coverage as a result of adsorbent–adsorbate interactions. Additionally, the model assumes that adsorption is characterized by a uniform distribution of binding energies up to maximum binding energy [96].

The Tempkin isotherm was calculated from the following form [97]:q_e_ = B Ln A + B Ln C_e_(10)
q_e_ = B Ln (AT Ce)(11)
where A is the constant of equilibrium binding (L g^−1^) associated with higher binding energy, B = (RT/b) (J mol^−1^) is the Tempkin constant and is related to the heat of adsorption and the gas constant (R = 8.314 J mol^−1^ K^−1^), and b (J mol^−1^) is a constant corresponding to the heat of sorption.

A plot of q_e_ vs. Ln C_e_ enables the determination of isotherm constants A and b_T_. Table 4 shows the values of the parameters. The data revealed that the Tempkin isotherm model applies to the CVD adsorption onto *S. costatum* adsorbent, as shown by the low regression correlation coefficient (R^2^ = 0.895). This can be obvious when the estimated q_e_ values differ from those derived from the experimental study, as in Figure 12.

#### 3.4.4. Halsey and Henderson Isotherm

These models are suitable for heterosporous solids and the multilayer sorption technique [98]. The Halsey model was calculated using the following equation [99]:(12)$$\mathrm{Ln}\;q_{e}=\frac{1}{n}\mathrm{Ln}\;K+\frac{1}{n}\;\mathrm{Ln}\;C_{e}$$

Meanwhile, the Henderson model was obtained from the following equation [97]:(13)$$\mathrm{Ln}\;[-\mathrm{Ln}(1-C_{e})]=\mathrm{Ln}\;K+\frac{1}{n}\;\mathrm{Ln}\;q_{e}$$
where n and K are Halsey constants for both Halsey and Henderson isotherms. A plot of ln q_e_ versus ln C_e_ for Halsey and ln [−ln(1−C_e_)] versus ln q_e_ for Henderson. The isotherm constants and correlation coefficients are summarized in Table 4. Halsey showed a correlation coefficient of R^2^ = 0.936 (Figure 13), while Henderson showed an R^2^ of 0.780 (Figure 14). The results obtained display that the Halsey model is applicable for the adsorption of CVD onto the adsorbent.

#### 3.4.5. Error Analyses

This study used isotherm models to evaluate the materials’ removal effectiveness, and the linear correlation coefficient was used to determine the relationship between the experimental data and the model under study (R^2^). The Chi-square (X^2^) test was additionally used as a standard to ensure the fitting quality was the non-linear regression. This statistical analysis is based on the sum of the squares of the differences between experimentally determined data and model-calculated data, where each squared difference was divided by the corresponding data produced by modeling, as calculated by the following Equation (14).
(14)$$X^{2}=\frac{{{(q}_{e,\mathrm{isotherm}}{-q}_{e,\mathrm{calc}})}^{2}}{q_{e,\mathrm{isotherm}}}$$

Based on the coefficient of determination (R^2^) value and the smallest Chi-square test (X^2^) result, the best fit model for the adsorption isotherm was selected. Halsey and Freundlich isotherm equations were found to best match the equilibrium data acquired for CVD onto *S. costatum* after analysis of R^2^ values and the data were fitted. When compared to the other isotherm forms, the Langmuir, Henderson, and Tempkin forms provided the worst fit to the experimental data. Additionally, the Chi-square test of both the Halsey and Freundlich isotherm yielded the lowest value, indicating that the q_e, cal_ of the Halsey and Freundlich isotherms and the q_e,_ exp_,_ as summarized in Table 4, are the isotherms that are most similar to one another. Therefore, with consideration of the R^2^ and the error functions, it is concluded that the Halsey and Freundlich isotherm best fitted the experimental data. The R_L_ and n_F_ values from the experiment indicate that the adsorption process is favorable and chemical.

### 3.5. Adsorption Kinetic

To examine the regulatory mechanism of the adsorption technique, for example, mass transfer and chemical reaction, Lagergren pseudo-first-order and pseudo-second-order kineticss in addition to the intraparticle diffusions were applied to the investigation of the experimental data of CVD adsorption by *S. costatum* in the batch examinations (Figure 11, Figure 12 and Figure 13).

#### 3.5.1. Pseudo-First–Order Kinetic

The pseudo-first-order rate expression of Lagergren depends on the sorption capacity of the adsorbent and is generally expressed as follows:Dq/d_t_ = K_1_ (q_e_ − q_t_)(15)
where q_e_ is the quantity of CVD adsorbed at equilibrium (mg g^−1^), q_t_ is the quantity of CVD adsorbed at time t (mg g^−1^), and K_1_ is expressed as a pseudo-first-order rate constant (min^−1^). The integrating equation was assessed [100] as follows:Log (q_e_/q_e_ − q_t_) = K_1_t/2.303(16)

The pseudo-first-order equation is given by the following formula in an equation:Log (q_e_ − q_t_) = Log q_e_ − K_1_t/2.303(17)

The plot of log (q_e_ – q_t_) agents t provides a straight line for the first-order adsorption kinetics. The value of the first-order rate constant k_1_ was given from the slope of the straight line. The K_1_ correlation coefficients, R^2^ values, and predicted and experimental q_e_ values for different *S. costatum*/CVD combinations are presented in Table 5. Figure 15 shows a plot of the form of the pseudo-first-order model at 25 °C. However, the correlation coefficients for the first-order kinetic model were low. In addition, the theoretical q_e_ values found from the first-order kinetic model did not give reasonable values (R^2^ = 0.752). This suggests that this adsorption process does not follow a first-order reaction.

#### 3.5.2. Pseudo-Second–Order Kinetic

The pseudo-second-order equations depend on the sorption efficiency of the solid phase, which, contrary to the other model, predicts the behavior over the whole range of concentrations. The pseudo-second-order kinetic rate equation is expressed as:dq_t_ /d_t_ = K_2_ (q_e_ − q_t_)^2^(18)
where K_2_ is the constant of the second-order rate (g m^−1^ min^−1^). The integrated equation is presented as follows:1/(q_e_ − q_t_) = 1/q_e_+ K_2_(19)

A form of the pseudo-second-order equation was obtained as described by Yang, et al. [101] as follows:t/q_t_ = 1/K_2_ q_e_^2^ + t/q_e_(20)

Plots of (t/q_t_) against (t) would provide a relationship (Figure 10), and the values of the q_e_ and K_2_ parameters can be calculated from the slope and intercept, respectively.

Meanwhile, the initial adsorption rate h_o_ (mg g^−1^ min^−1^) at the time was calculated by the following [102]:h_o_ = K_2_q_e_^2^(21)

The results indicate that the values of qe for second-order kinetics are closer to the estimated values (Table 5). The correlation coefficient (R^2^ = 0.978) value for the pseudo-second order is also closer to unity when compared to the first order (Figure 16 and Table 5). The near values between the predicted values of q_e_ from the pseudo-second-order model and the experimental q_e_, in contrast to the pseudo-first-order, predicted q_e_ values, which significantly differed from experimental q_e_, serve as more evidence for this. This shows that the adsorption process follows a pseudo-second-order model and that it appears to be governed by chemical adsorption, including valency forces, by sharing or exchange of electrons among dye anions and the adsorbent, which provides the best correlation of the data for the dye [73].

#### 3.5.3. Intra-Particle Diffusion Equation

Intra-particle diffusion is a key rate-controlling phase in the first phase of adsorption, and outside mass transfer or boundary layer diffusion could be defined by the initial rate of solute sorption. The acquired data were examined using mass transfer and kinetic models [103]. Adsorption kinetics are obtained by the following stages. The intra-particle diffusion model was calculated as follows [73,104]:q_t_ = K_dif_ t^1/2^ + C(22)
where C is the intercept and K_dif_ (mg g^−1^ min^−0.5^) reflects the intra-particle diffusion rate constant, which is estimated through the slope of the regression line. The intra-particle diffusion model showed no fit with the experimental data (R^2^ = 0.438). Moreover, the data assigned that, in the intra-particle diffusion plots for adsorption of CVD, it appeared that none of the regions has C values equal to zero, which did not pass throughout the origin (Figure 17), suggesting that the pore diffusion is not the rate-controlling step of the investigated adsorption process (Table 5).

### 3.6. Industrial Wastewater and Seawater Applications

A wastewater sample was collected from the El-Naboria drain, which belongs to Alexandria, Egypt, and comprises numerous industrial sewage and agricultural wastes. To assess the adsorbent validity of *S. costatum*, real wastewater mixed with simulated dye samples was collected to examine the removal of CVD by the adsorbent under optimal conditions (Figure 18). Owing to the low CVD concentrations, an amount of CVD was added to obtain 5 mg L^−1^ of this dye. The characteristics of the wastewater used are as follows: values of salinity (PPT) are 186.5 and pH is 8.44; the dissolved oxygen (mg L^−1^) value was 11.01, nitrite (135.45 μg L^−1^), nitrate (622.7 μg L^−1^), ammonia (468.5 μg L^−1^), and phosphate (52.37 μg L^−1^). The CVD solution was filtered to remove precipitates and suspended the matter. The presence of salt did not affect the removal of CVD by *S. costatum*, which leads us to deduce that there was no interaction between the salt and the *S. costatum*, nor between the salt and the CVD. Moreover, the high concentration of CVD ions may make them more preferably adsorbed by *S. costatum*, since the effect of salt cannot influence the active groups of the adsorbent. The results show that adding saltwater to the dye solution increases the adsorption capacity of the CVD, and the reason for this is that the interactions between solvent molecules and NaCl ions are fewer than that of dyes, which leads to an increase in the solubility of dyes in the aqueous solution, and thus leads to a desire in the ability to adsorb it on the surface. Furthermore, it is possible to explain this flow based on the process of imbibition. This is because this process leads to swelling of the adsorbent material and the removal of the osmotic pressure of the saline solution, which leads to a change in the geometry of the pores of the adsorbent material, and thus leads to a decrease or increase in adsorption according to changes of the shape of the surface [105]. The application of *S. costatum* to remove the CVD from wastewater was carried out under optimized conditions (pH 3 for 180 min at 25 °C and 0.05 g of adsorbents in the solution). Deionized water comprising a similar concentration of CVD was applied as a control to estimate the effect of *S. costatum* on CVD. The obtained results reveal that changing the type of wastewater had a substantial impact on the greatest removal, with deionized water having the least impact on the adsorption method, with a color removal of 90.9% at pH 3 after 180 min.

By contrast, real wastewater comprises very high concentrations of interfering ions from numerous pollutants, which had a significant influence on the elimination efficiency of CVD, with a color removal of 10.08 % after 180 min at pH 3. Regardless, the results reveal that *S. costatum* adsorbents can be successfully used as a low-cost adsorbent to remove CVD from aqueous mixtures and wastewater. Furthermore, changing the same kind of CVD solution did not affect the maximal adsorption capabilities.

### 3.7. Regeneration and Reusability Study

The regeneration and reuse of the sorbent material are essential for its economic viability [8]. To evaluate the reusability of *S. costatum*, adsorption–desorption experiments were conducted using 5 mg L^−1^ of CVD solution, shaken at 110 rpm for 3 h, as presented in Figure 19. The current results reveal that the removal efficiency of the *S. costatum* for CVD uptake remained nearly unchanged for the three consecutive series. The initial removal efficiency was 90.10% in 180 min, which was reduced to 69.30% after the first round. Hence, it could be concluded that the sorption capacity of the adsorbent remains unaffected with extended regeneration rounds. This decrease shows that the adsorbent represents a good material that can be used in industrial operations. It can be easily known that desorption efficiency decreases with increasing cycle numbers due to the decrease of adsorption capacity. This indicates that, while biomass-adsorbing sites were still available, they became more difficult to locate. Changes in the chemistry and structure of the adsorbent, as well as changes in the mass transport conditions, may have an influence on adsorption performance after the third cycle of use [106].

### 3.8. Comparison with Other Studies

Table 6 presents the comparison amongst the maximum values of CVD adsorption capacities of the prepared adsorbent, and some adsorbents, comprising some activated carbons, modified biomasses, nanoparticles, and zeolite. In the present work, the amount of maximum monolayer uptake capacity of CVD by *S. costatum* was obtained at 6.41 mg g^−1^; this amount has been compared with q_m_ achieved from other studies on the removal of CVD. Table 6 shows the maximum uptake capacity from the Langmuir model by various sorbents.

### 3.9. Suggested Mechanism

The adsorption mechanism could be affected by a wide variety of factors. Adsorption is the outcome of the interaction between ions, molecules, and the surface of the adsorbent. The surface of the adsorbent carries different kinds of functional groups, such as amino, carbonyl, phosphate, carboxylate, carboxyl, and hydroxyl, as reported by FTIR analysis, which are responsible for the confiscation of harmful materials from industrial wastewater [119]. However, the dye remediation process is mainly comprised of two mechanisms: active uptake and passive uptake [65]. The active uptake is the exchange of monovalent and divalent ions such as Na, Mg, and Ca in the cell walls, which are replaced by ions of dyes. Passive uptake is the formation of a complex between the ions of dyes with functional groups located on the cell wall; this process is alternating and fast, while active uptake is a process that occurs when dye ions bind to the cell [65]. This mechanism occurs in combination with the usage of dye ions by microorganisms and intracellular accumulation of dye ions; dyes can also be deposited in the metabolism and excretion process on a second level. pH and the presence of other ions are factors that influence the uptake passive process. pH and the presence of other ions influence the passive uptake process, whereas pH and temperature influence the active uptake process [120].

On the other hand, Soedarti et al. [66] found that *Skeletonema* sp., including microalgae, are aquatic organisms containing molecular mechanisms that may discriminate non-essential toxic metals from metal ions that are required for growth. *Skeletonema costatum* can absorb CVD ions in two ways: absorption and adsorption, according to this study. *S. costatum* has cell walls, which allow for adsorption. *S. costatum* has cellulose-based cell walls. The functional groups in the cellulose inside cell walls, such as hydroxyl, can bond with ions in dye [66]. *S. costatum* also absorbs nutrients because it produces phytochelatins [59]. *S. costatum is* like other microalgae that also produces phytochelatins, namely, peptides metallothionein class III (Mt-III), to detoxify heavy metals, which can be induced by the existence of heavy metal ions, as reported by Perales-Vela et al. [121] and Shaw [122].

## 4. Conclusions

The diatom *Skeletonema costatum* was found to be effective in the uptake of Crystal Violet Dye (CVD). Maximum dye adsorption was achieved by a batch technique at pH 3, adsorbent dose 0.4 g, contact time 120 min, and 5 mg L^−1^ initial dye concentration at 25 °C. The percentage removal of the *S. costatum* diatom rose to 95.2% for an adsorbent dosage of 0.4 g. Freundlich and Halsey’s isotherm models describe the adsorption of CVD onto *S. costatum*, suggesting that adsorption was onto a non-uniform site and that the CVD adsorption data followed the pseudo-second-order kinetic model. Thermodynamic data indicate the endothermic nature of adsorption systems. The calculated R_L_ value was less than 1, while the value of n was greater than 1, hence the adsorption process was favorable. Values of q_e_ agreed with the values of the current experiment. However, initial rapid adsorption was onto a uniform site, thus maximum monolayer adsorption capacity was obtained to be 6.410 mg g^−1^. The *S. costatum* displayed significant regeneration possibility and application on real wastewater. Finally, the current work concluded that the marine diatom *S. costatum* is significantly used to eliminate the textile toxic dye from wastewater, since it is an effective, eco-friendly, and locally available adsorbent.

## Figures and Tables

**Figure 1 materials-15-06375-f001:**
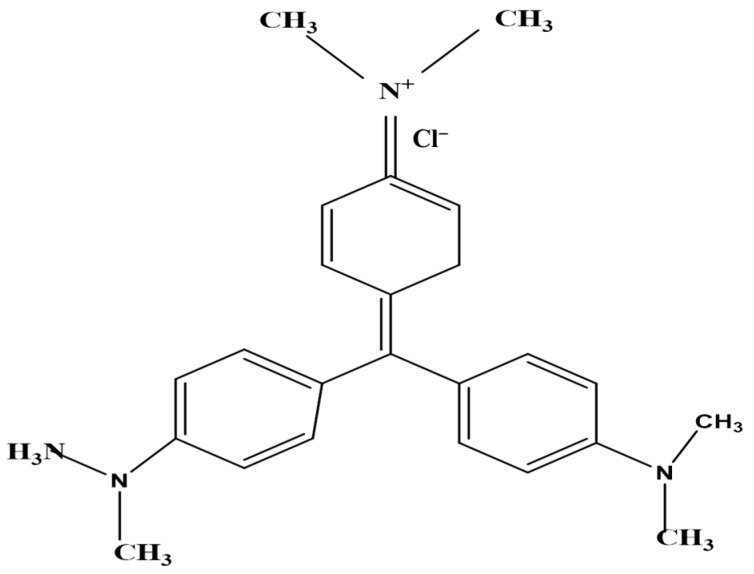
The chemical structure of the crystal violet dye (CVD).

**Figure 2 materials-15-06375-f002:**
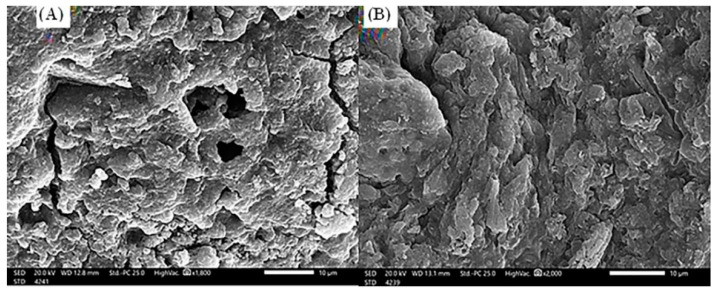
Scanning electron microscope (20.0 kv, ×2000) of *S. costatum* adsorbent before (**A**) and after (**B**) adsorption of CVD.

**Figure 3 materials-15-06375-f003:**
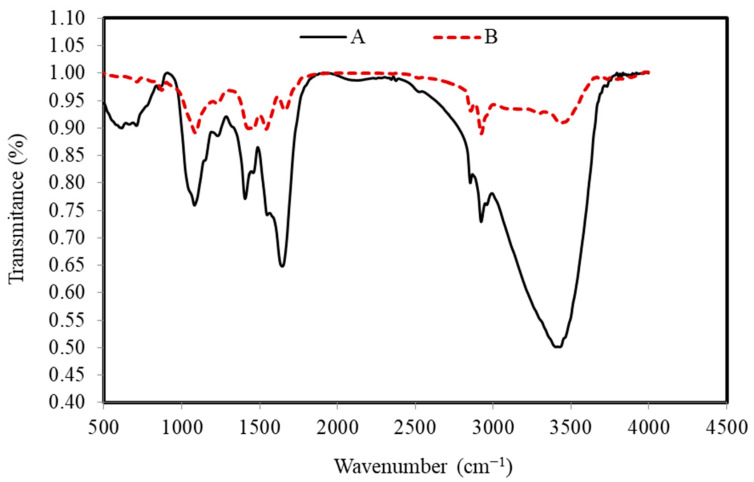
FTIR peaks of the *S. costatum* adsorbent before (A) and after (B) adsorption of CVD.

**Figure 4 materials-15-06375-f004:**
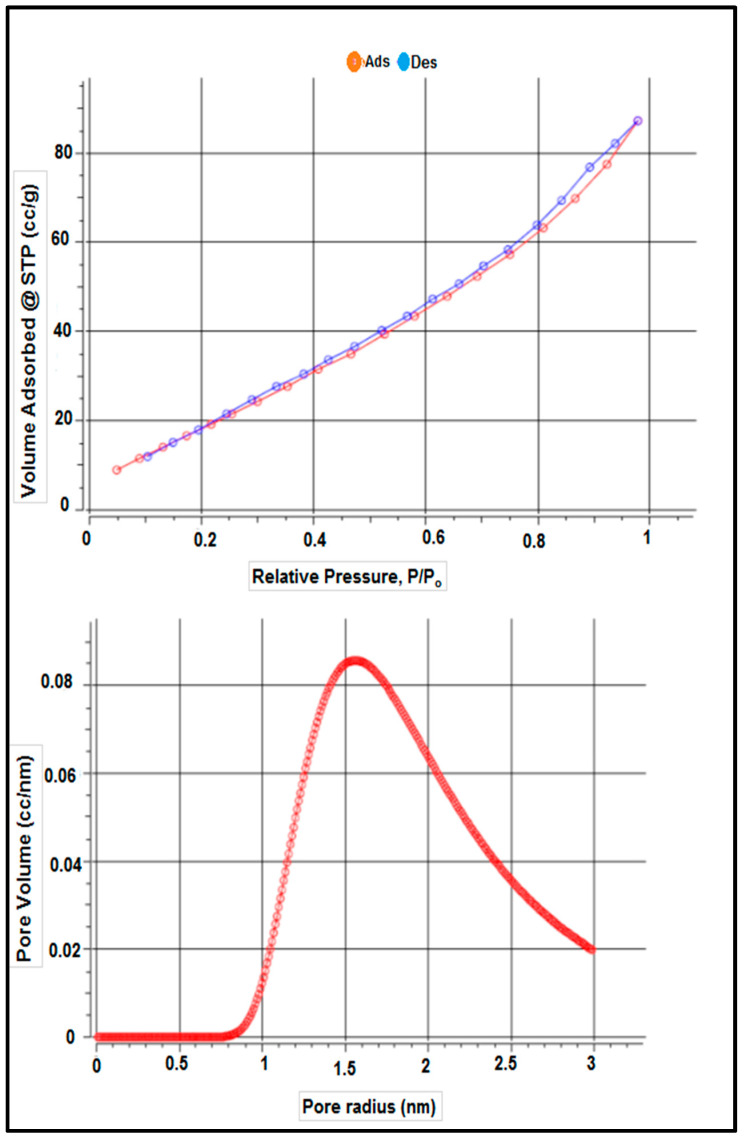
Isotherm–Isotherm isotherms and the pore size distribution.

**Figure 5 materials-15-06375-f005:**
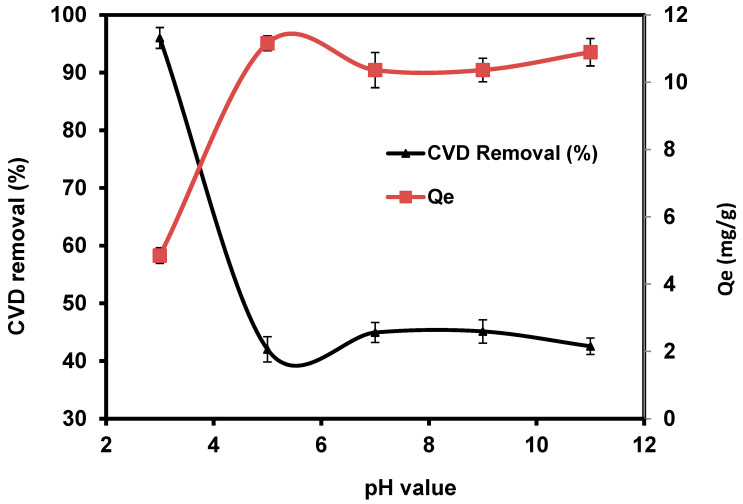
Impact of pH values on the adsorption of CVD onto *S. costatum*.

**Figure 6 materials-15-06375-f006:**
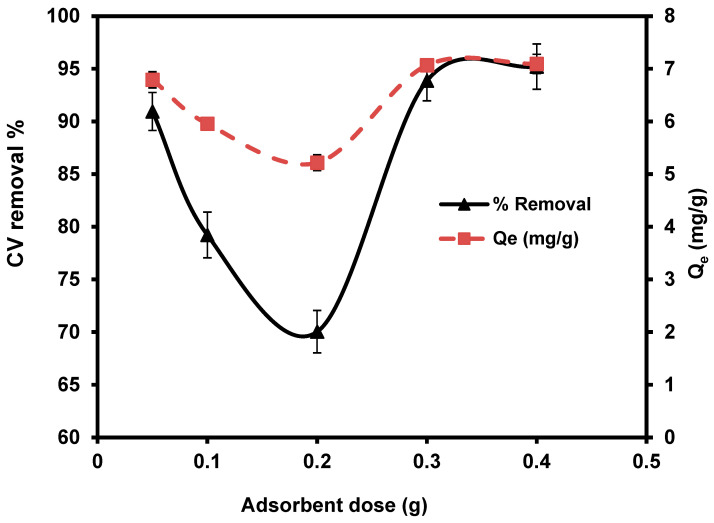
Impact of *S. costatum* concentrations on the adsorption of CVD.

**Figure 7 materials-15-06375-f007:**
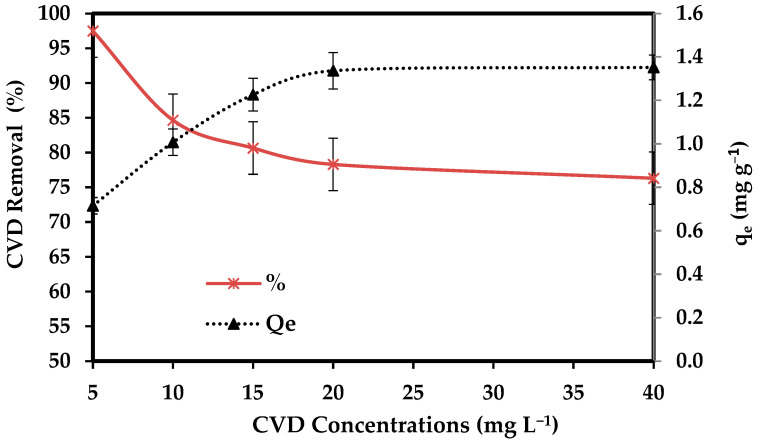
Impact of CVD concentrations on the adsorption process.

**Figure 8 materials-15-06375-f008:**
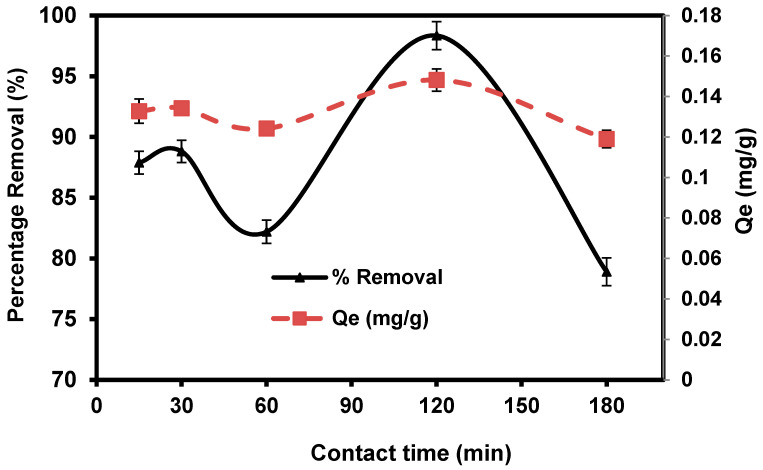
Impact of agitating time in the percentage removal of crystal violet dye.

**Figure 9 materials-15-06375-f009:**
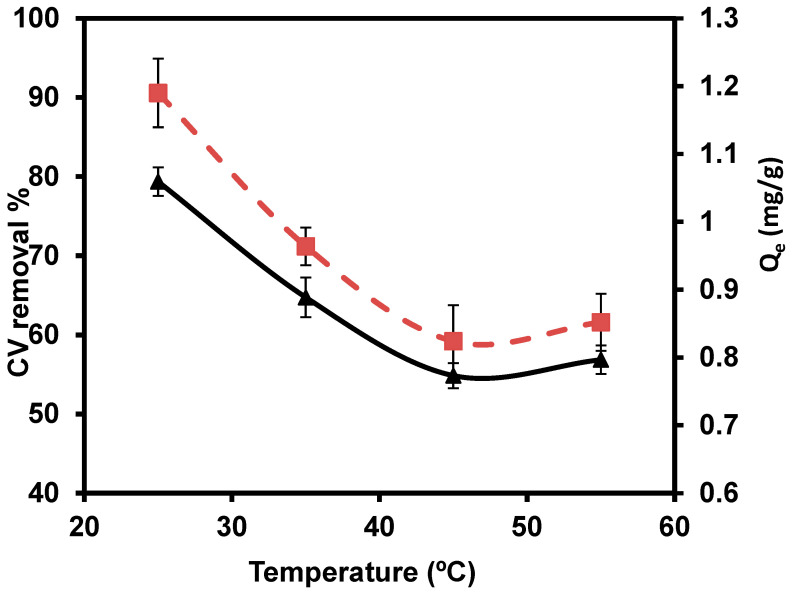
Impact of several temperatures on the removal percentage of CVD.

**Figure 10 materials-15-06375-f010:**
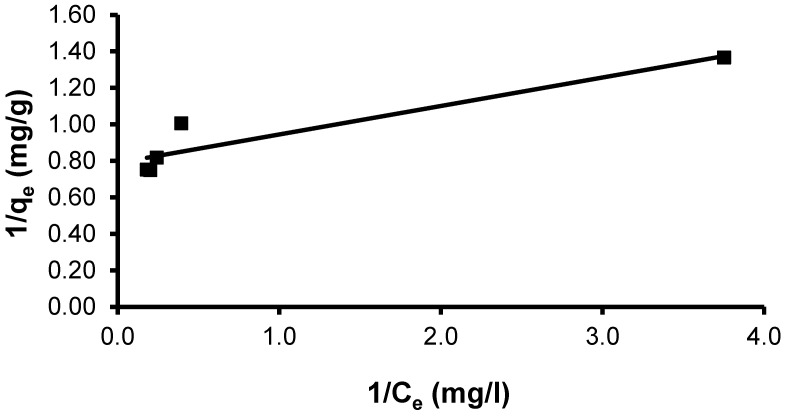
Plot of the Langmuir isotherm model for CVD adsorbed onto *S. costatum*.

**Figure 11 materials-15-06375-f011:**
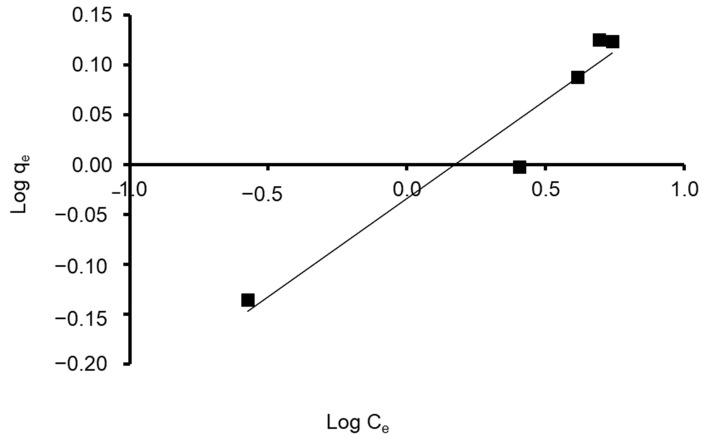
Plot of the Freundlich isotherm model for CVD adsorbed onto *S. costatum*.

**Figure 12 materials-15-06375-f012:**
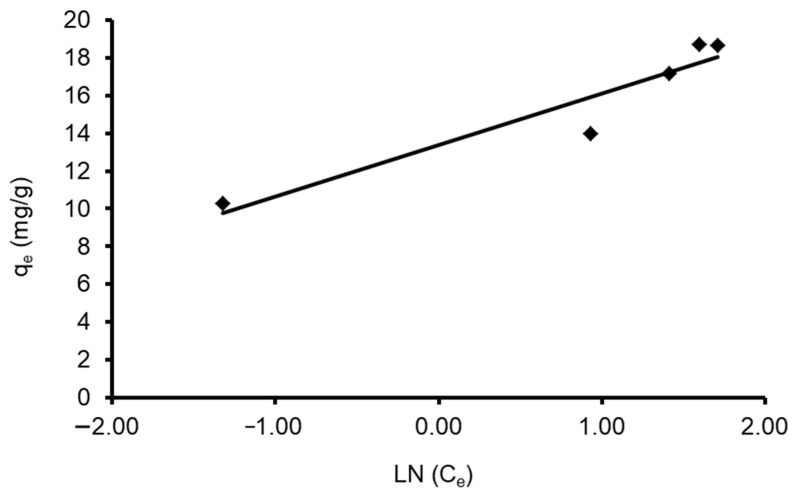
Plot of the Tempkin isotherm models for CVD adsorbed onto *S. costatum*.

**Figure 13 materials-15-06375-f013:**
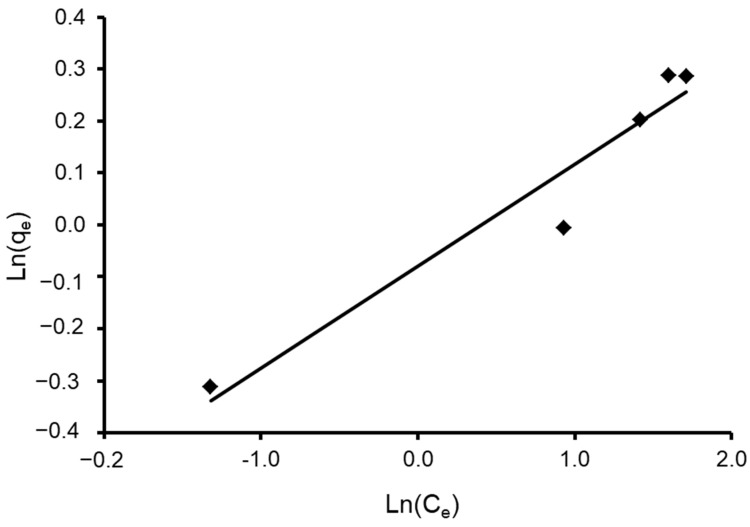
Plot of the Halsey isotherm model for CVD adsorbed onto *S. costatum*.

**Figure 14 materials-15-06375-f014:**
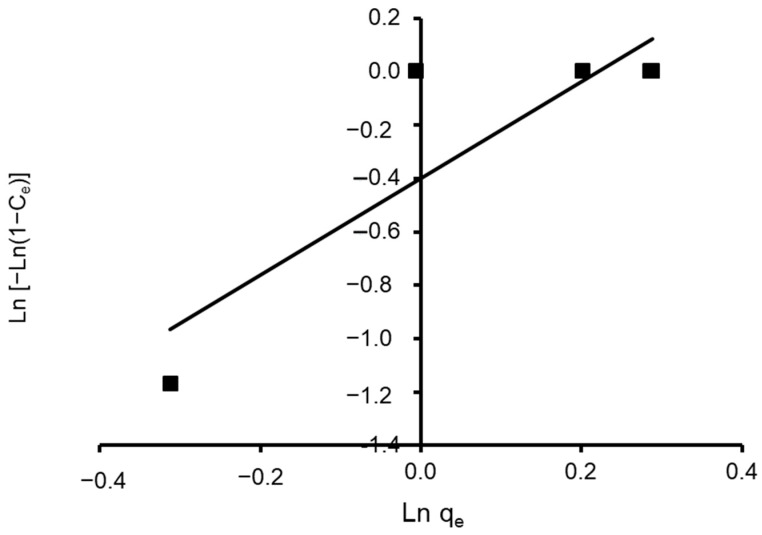
Plot of the Henderson isotherm model for CVD adsorbed onto *S. costatum*.

**Figure 15 materials-15-06375-f015:**
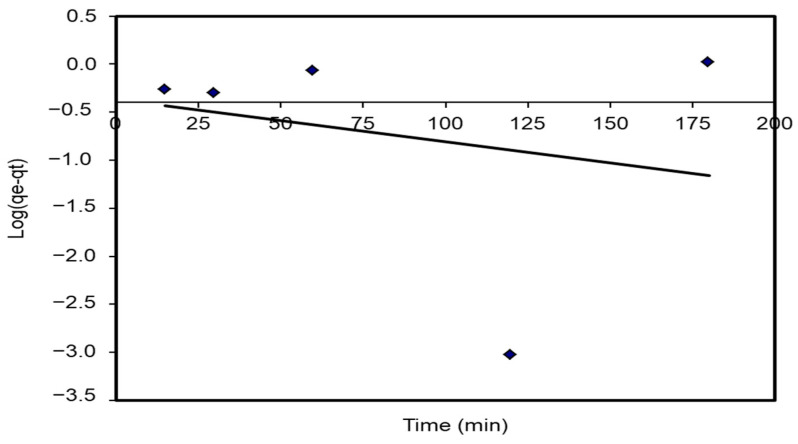
Pseudo-first-order plots for the adsorption of CVD onto *S. costatum*.

**Figure 16 materials-15-06375-f016:**
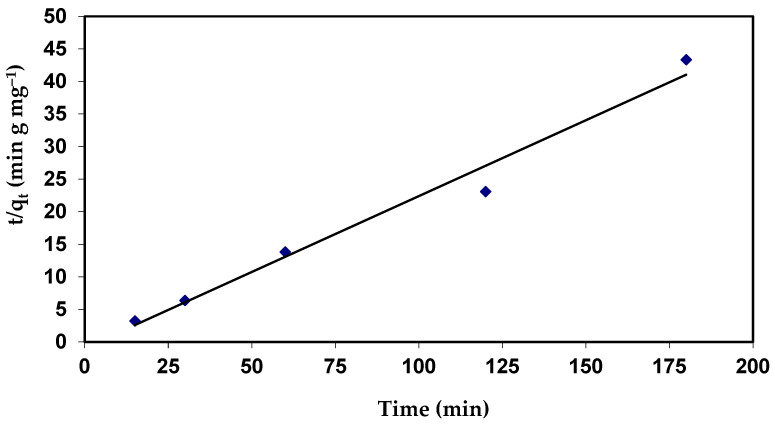
Pseudo-second-order plots for the adsorption of CVD onto *S. costatum*.

**Figure 17 materials-15-06375-f017:**
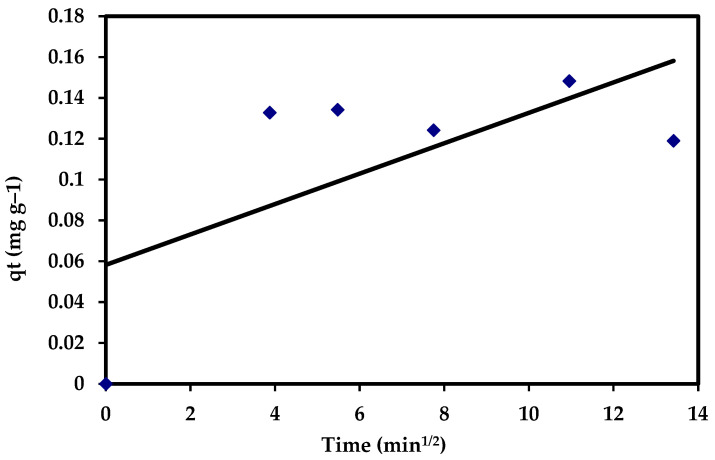
The intra-particle diffusion plots for the adsorption of CVD onto *S. costatum*.

**Figure 18 materials-15-06375-f018:**
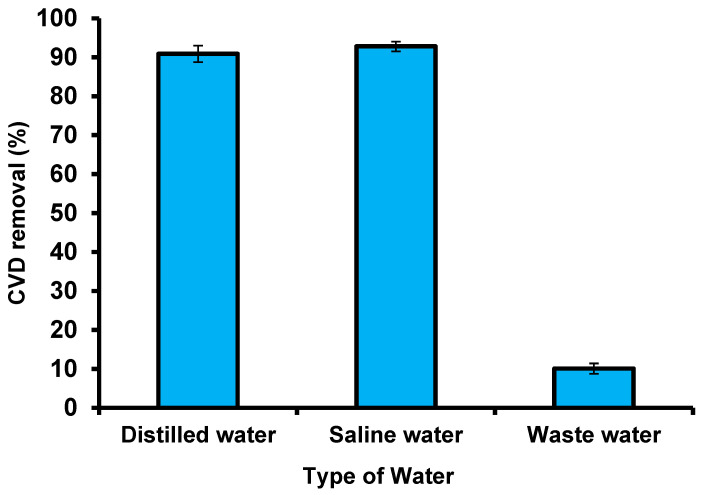
Removal of CVD from different solutions using *S. costatum*.

**Figure 19 materials-15-06375-f019:**
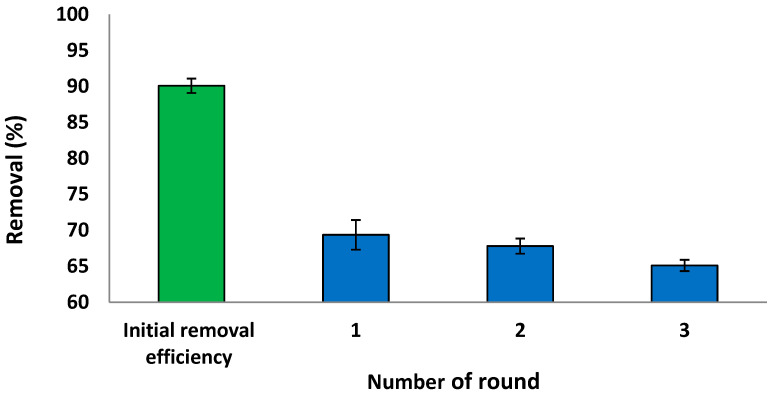
Reusability study of the *S. costatum* after adsorption of CVD.

**Table 1 materials-15-06375-t001:** Physical properties of *S. costatum*.

Properties	Data
BET surface area	87.17 m² g^−1^
Single-point BET	74.10 m² g^−1^
DH desorption	62.89 m² g^−1^
Skeletal density	2.2 g C^−1^
Average pore sizes	3.131 nm
Average pore volume	0.103 CC g^−1^

**Table 2 materials-15-06375-t002:** The calculated values of thermodynamic parameters of the adsorption of CVD onto *S. costatum*.

T (°C)	Q_e_ (mg g^−1^)	ΔG (kJ mol^−1^)	ΔH (kJ mol^−1^)	ΔS (kJ mol^−1^ k)
25	1.190	−5.050	8.636	−0.036
35	0.964	−3.279		
45	0.824	−2.355		
55	0.852	−2.636		

**Table 3 materials-15-06375-t003:** Relationship between RL and the type of isotherm [92].

R_L_	Type of Isotherm
**R_L_ > 1**	Unfavorable
**R_L_ = 1**	Linear
**0 < R_L_ < 1**	Favorable
**R_L_ = 0**	Irreversible

**Table 4 materials-15-06375-t004:** Factors of the isotherm models from linear solvation.

Isotherm Model	Isotherm Parameters	Values
**Langmuir**	Qm (mg g^−1^)	17.76
	R_L_	0.011
	K_L_	5.06
	R^2^	0.876
	X^2^	1.013
**Freundlich**	1/n	0.196
	nF	5.102
	K_F_ (mg^1–1/n^ L^1/n^ g^−1^)	12.94
	R^2^	0.936
	X^2^	0.001
**Tempkin**	A_T_	136.38
	B_T_	13.38
	b_T_	185.16
	R^2^	0.895
	X^2^	1205.66
**Halsey**	n	5.1
	K	475,898
	R^2^	0.936
	X^2^	0.002
**Henderson**	n_h_	1
	K_h_	0.006
	R^2^	0.78
	X^2^	62.093

**Table 5 materials-15-06375-t005:** The calculated values of kinetic parameters of the adsorption of CVD onto *S. costatum*.

Kinetic Models	Parameters	Values
The pseudo-first-order	q_e_ (calc.) (mg g^−1^)	63.09
	k_1_ (min^−1^)	0.00414
	R^2^	0.752
	q_e_ (exp.) (mg g^−1^)	0.1316
The pseudo-second-order	q_e_ (calc.) (mg g^−1^)	0.12
	k_2_ (mg g^−1^ min^−1^)	2.16
	h_o_	0.03
	R^2^	0.978
	q_e_ (exp.) (mg g^−1^)	0.1316
The intraparticle diffusion	K_dif_ (mg g^−1^ min^−0.5^)	0.0583
	C (calc.) (mg g^−1^)	0.01
	R^2^	0.438

**Table 6 materials-15-06375-t006:** The comparisons of the maximum uptake capacity of CVD by different adsorbents.

Material	q_e_ (mg g^−1^)	Refs.
Grapefruit peel	254.16	[22]
Lignin-rich Isolate from Elephant Grass	24.99	[102]
Pod-inspired MXene/porous carbon microspheres	750.00	[107]
Fe_3_O_4_ nanoparticles	269.70 to 282.50	[107]
Graphene oxide cross-linked hydrogel nanocomposites of xanthan gum	371.29	[108]
Soil-Ag nanoparticles	1.92	[109]
Cellulose-based biosorbent	153.85	[110]
Lemongrass leaf fibers incorporated with cellulose acetate	36.10	[111]
Unused solid waste of Rosewater extraction	168.8	[112]
Chitin	420.10	[113]
Zeolite from bottom ash	17.60	[114]
Natural zeolite	106.67	[115]
Magnetic Zeolite	0.97	[116]
Leaf biomass of *Calotropis procera*	4.14	[117]
Blue-green alga, *Spirulina* sp.	101.87	[118]
Marine diatom, *Skeletonema costatum*	6.41	Present study

## Data Availability

Not applicable.

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
