# Peer review of "Equilibrium and Kinetic Modeling of Crystal Violet Dye Adsorption by a Marine Diatom, Skeletonema costatum"

_materials, 2022, doi:10.3390/ma15186375_

Round 1
Reviewer 1 Report
The work of Mohamed Ashour and his co-workers was carried out in order to study the equilibrium and kinetic modeling of Crystal Violet dye adsorption by Marine Diatom, Skeletonema costatum.
My observations are listed in the attached file.

Author Response
The authors would like to thank Reviewer # 1 for his/her valuable comments which significantly improved the manuscript. The reviewer comments have been considered carefully by the authors’ responses. All corrections were conducted as the Reviewer suggested in the manuscript in Red.

Reviewer 2 Report
In this manuscript, the authors studied the effect of pH, contact time, adsorbent dosage, temperature, and initial crystal violet dye concentration in a separate adsorption batch reaction. The adsorbents exhibited excellent adsorption behavior for removal of CVD from aqueous solution. On the whole, the findings are of considerable interest and well done. I recommend it to be published after a major revision.
1. Maybe the author should compare their results clearly with other reported works containing different alga, highlighting the advantage and disadvantages of their novel composite.
2. The authors are responsible for the English, which should be polished throughout the manuscript to clear some minor typo/grammar errors.
3. Introduction part, if possible, some important and relative reports about self-assembled nanostructures from various styles (https://doi.org/10.1016/j.colsurfa.2021.127261)
4. All equation should be revised, which contain some typo error.
5. The author should better improve the beauty and quality of the figures in the manuscript.
6. In cyclic stability, why the dye removal decreases after first round from 90 to 70 compared the small variation after the 1st round.
7. error bars in all necessary figures should be added.
8. nitrogen sorption isotherms should be included in the manuscript and the pore size distribution curves and the authors should also identify the isotherm type.
Hence, I recommend it accepted for publication after some major revisions.
Author Response
The authors would like to thank Reviewer # 2 for his/her valuable comments which significantly improved the manuscript. The reviewer comments have been considered carefully by the authors’ responses. All corrections were conducted as the Reviewer suggested in the manuscript in Yellow.

Round 2
Reviewer 1 Report
Dear authors,
I have some comments:
-Please correct the paragraph from L390-393. ''The data were fitted to the Langmuir, Freundlich, Tempkin, Halsey, and Henderson isotherm equations and the fixed parameters were calculated (Table 4).''
-Please remove Figure 10. The representation of qe vs. Ce shows the nonlinear fit of isotherm models and in the manuscript there are present the factors of the isotherm models from linear solvation (Line 494);
-Line 591 and Line 595: Please check the numbering of the figure
-Line 594: Time1/2 (min1/2) instead of Time1/2 (minutes)
-Table 5: Please check the value f k1 parameter;
-Line 617 and Line 644: Please check the numbering of the figure;
-Line 669: Please check the numbering of the figure;
Author Response
Dear Reviewer, We would like to thank you for your professionally reviewing. All comments were answered carefully. Thank you for your efforts. Authors, |

Reviewer 2 Report
The manuscript is acceptable in the present form since authors have declared all the questions
Author Response
Dear Reviewer, We would like to thank you for your professionally reviewing and efforts Authors, |